# Confirmation of the local establishment of alien invasive turtle, *Pseudemys peninsularis*, in South Korea, using eggshell DNA

**Seung-Ju Cheon**[1◉]**, Md. Mizanur Rahman**[2◉]**, Ji-A Lee**[1]**, Seung-Min Park**[1]**, Jae-Hong Park**[1]**, Dong-Hyun Lee**[2]**, Ha-Cheol Sung**[2]*

**1** Department of Biological Sciences and Biotechnology, Chonnam National University, Gwangju, South Korea, **2** Department of Biological Sciences, Chonnam National University, Gwangju, South Korea

◉ These authors contributed equally to this work.
* shcol2002@jnu.ac.kr

## Abstract

Alien invasive species are posing conservation challenges worldwide. Pet trade, one of the many ways, is worsening the situation. Especially, pet turtles have been released into nature due to their longer life span and peoples' religious and traditional beliefs. In addition, unwanted and undesired pets are also released. While information on the successful local establishment and subsequent dispersal into new habitats is required to designate an invasive and ecosystem-disturbing species, alien freshwater turtle nests have always been hard to find and identify in nature. Because one should identify nests by the eggs, which do not always guide properly, as adults abandon the sites quickly. We thought the recent advancement in DNA technology may help improve the situation. We studied *Pseudemys peninsularis*, one of the most traded freshwater turtle pet species, which has already been reported from a wide range of wild areas in South Korea. Yet, it is not designated as ecosystem-disturbing species due to a lack of adequate information on their local reproduction and establishment. We conducted surveys and found two nests in Jeonpyeongje Neighborhood Park, Maewol-dong, Seo-gu, Gwangju. We developed the methodology for extracting DNA from the eggshells and successfully identified the nests by phylogenetic analysis and verified through egg characteristics and morphological features of artificially hatched juveniles. This was the first successful initiative to extract DNA from freshwater turtle eggshells. We believe it will help future researchers identify the alien invasive turtle nests and develop their control and management policies. In addition, our study also included comparative descriptions and schematic diagrams of the eggs of eight freshwater turtles, including a native and three ecosystem-disturbing species, from South Korea. We urged an immediate designation of *P. peninsularis* as an ecosystem-disturbing species considering its local establishment, distribution range, and potential negative impact on native ecosystems.

**Data Availability Statement:** All relevant data are within the paper.

**Funding:** H-CS: RE201807039; Korea Environment Industry & Technology Institute, funded by the

Korea Ministry of Environment (MOE). The funders had no role in study design, data collection and analysis, decision to publish, or preparation of the manuscript.

**Competing interests:** The authors have declared that no competing interests exist.

## Introduction

In the era of conservation challenges, invasive species is a growing global concern [1, 2]. Among many other ways, the pet trade is accelerating the spread of invasive species around the world [3]. Although many countries formulated strategies to control the spread of alien invasive species, unfortunately, a few of the laws in some countries help increase the events of undesired introductions of pet species into the wild [4 and references therein]. For instance, Brazil, Chile, Indonesia, and Isarel do not prohibit the import of fish and other aquatic species [4]. However, the countries like South Korea, which are already well-known as pet trade epicenters, are also at high risk of getting invasive species into their nature despite having many strict laws [5]. To date, species from Testudines are the most frequently wild recorded alien species among 677 reported herpetofauna (110 are turtles) from pet shops in Korea [6]. The report of alien turtle species in the nature of Korea mainly resulted from the abandonment by the pet owners and release during religious events [7, 8]. Currently, more than 13 alien turtle species are currently found in the wild [9, 10]. To facilitate the management and have a legal ground to control, *Trachemys* spp., *Pseudemys concinna*, *Mauremys sinensis*, *Macrochelys temminckii*, *Pseudemys nelsoni*, and *Chelydra serpentina*, among the wild reported alien turtles, have been designated as ecosystem-disturbing species [11].

However, it is necessary to ensure the successful local establishment and subsequent dispersal and secondary spread into new habitats after the introduction of a species to designate it as an invasive and ecosystem-disturbing species [12]. In the case of reptiles, a species is considered to be established if there is evidence of maintaining a population by laying eggs in the local environment after getting released, abandoned, or escaped [13, 14]. Unfortunately, we have little information on the nest sites and eggs of most of the alien turtle species, including *Pseudemys peninsularis*, recorded from the wild of South Korea.

Freshwater turtle nests have always been hard to find in nature [15]. Furthermore, the identification of turtle nests is largely depended on the identification of the nest-building and egg-laying individuals [16], while some species take a very little amount of time for it [17]. Thus, few researchers suggested using egg external features for turtle nest identification [18, 19], whereas others argued it was ineffective due to overlapping egg diameters [20]. Like many other animal groups, recent studies showed the successful use of DNA barcoding in identifying herpetofauna [21–23]. Hence, we hypothesized that eggshells can be used as a source of DNA and in identifying freshwater turtle nests and species, including *P. peninsularis*, which is already well-practiced in birds and sea turtles [24–26].

We studied *P. peninsularis*, one of the most traded freshwater turtle pet species in South Korea which has already been reported from a wide range of wild areas [27]. Yet, it is not designated as ecosystem-disturbing species because of a lack of adequate information on the successful reproduction and secondary spread in the local environment. Considering the absence of information on the nests and eggs in nature, confirmation is needed for the establishment of this species in the native ecosystems. Thus, we conducted surveys to discover *P. peninsularis* nests and eggs in the wild of South Korea. We developed a method for extracting genomic DNA from freshwater turtle eggshells and used partial mitochondrial DNA sequence for molecular identification of the species. This is the first initiative for extracting DNA from freshwater turtle eggshells and using it in species identification. Thus, we measured the egg characteristics and checked the morphology of the hatchlings after hatched the eggs artificially to verify the molecular-based identification and validate the sequences we amplified from the eggshells. Furthermore, we presented the comparison among egg characteristics and external features of Korea inhabiting eight turtle species, including a native and three ecosystem-disturbing species. Finally, we suggested designating

*P. peninsularis* as an ecosystem-disturbing species in Korea and strengthening its control and management policies.

## Material and method

### Discovery of nests and egg collection

Considering previous reports of many alien turtles, the Jeonpyeongje Neighborhood Park, Maewol-dong, Seo-gu, Gwangju (35˚ 6'55.14"N 126˚50'54.24"E; Fig 1), was surveyed for the observation of possible nest sites of *P. peninsularis*. The Jeonpyeongje Neighborhood Park, has an area of 4.699 ha. It has a reservoir, which is easily accessible by a wooden bridge and is reported to contain many alien turtles, including *T. scripta elegans*, *T. scripta troostii*, *P. concinna*, and *P. peninsularis*. Thus, we surveyed the reservoir searching for the *P. peninsularis* nests. We recorded the external features and measured the eggs for comparing and identifying the species. The damaged eggs were excluded from the measurements. The eggs were collected for further analysis.

### Identification and comparison of eggs

To confirm the identity of the species and the nest, we measured and compared the egg parameters with previous literature. We followed Congdon and Gibbons (1985) and Iverson and Ewert (1991) [18, 28]. We recorded the clutch size of the nests, shell type, mass, length, width, and elongation of the eggs.

### DNA extraction

We used DNeasy Blood & Tissue kit (QIAGEN, Germany) and followed a partially modified manufacturer's protocol for DNA extraction. First, we took 60 mg of egg membrane from the eggshell and mixed it with ATL buffer 200μl, proteinase K 20μl, and dithiothreitol (DTT; 1mg/ml) 9.8μl to react 24h at 55˚C. After centrifugation at 18,000xg for a minute, the supernatant was transferred to a new 1.5 ml tube and mixed with AL buffer 200μl and ethanol (95–100%) 200μl. Then the samples were transferred to the Qiagen column and centrifuged again at 6,000xg for a minute. After that, we added AW1 buffer 500μl to the column and again centrifuged at 6,000xg for a minute. In the next step, we added AW2 buffer 500μl to the column and centrifuged at 20,000xg for 3 minutes. After the column was transferred to a new 1.5ml tube and eluted in 70μl and centrifuged at 6,000xg for a minute. Polymerase chain reaction (PCR) amplifications were performed using Solg Pfu DNA polymerase (Solgent, Korea) to confirm the mitochondrial DNA sequence. We amplified a Control region, a widely used species-specific sequence, using DES_1 (5'-GCATTCATCTATTTTCCGTTAGCA-3')/DES_2 (5'-GGATTTAGGGGTTTGACGAG-3') [29]. Newly generated sequences in the present study were deposited in GenBank (Table 1).

### Phylogenetic analysis

To confirm the identity of the specimens via DNA barcoding and to evaluate their matrilineal relationships, we analyzed our new sequences together with relevant homologous sequences in GenBank (Table 1). Based on previous phylogenetic analyses [30] we used *Chrysemys picta* as the outgroup. All sequences were aligned by MEGA X [31]. We inferred matrilineal genealogy using Bayesian Inference (BI) analyses and Maximum Likelihood (ML) methods. We used MrBayes 3.1.2 [32] for BI analyses, while the best model was assessed with the software JMODELTEST [33]. The GTR+Gamma was found to be the best model. The analyses were conducted with 10 million generations and sampled every 1000

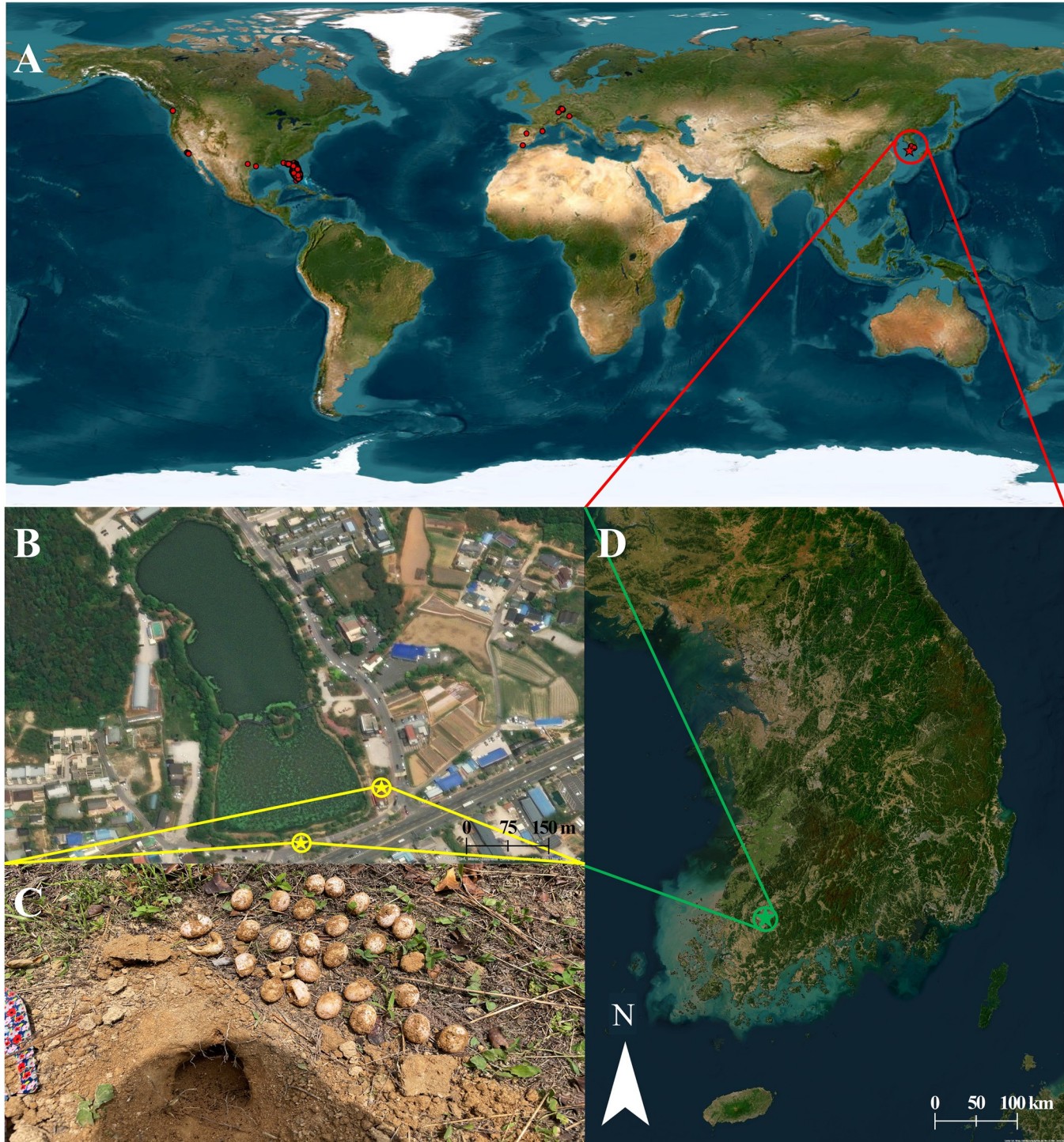

**Fig 1. Distribution of *Pseudemys peninsularis* and the study area.** (A) Distribution of *P. peninsularis* across the world. The red star indicates the location of South Korea. (B) Jeonpyeongje Neighborhood Park with the nest sites indicated by the yellow stars. (C) A nest with *P. peninsularis* eggs. (D) Map of South Korea with study site at Gwangju indicated by a green star. The maps and satellite images were generated using Arcmap 10.8.1 (ESRI, USA; https://support.esri.com/en/products/desktop/arcgis-desktop/arcmap/10-8-1).

**Table 1. List of sequences used in the present study.**

| Serial No. | Description | voucher | region | Accession number | Locality | Reference |
|---|---|---|---|---|---|---|
| 1 | *Pseudemys peninsularis* | | **D-loop** | **OQ186456** | **South Korea** | **Present study** |
| 2 | **P. peninsularis** | | **D-loop** | **OQ186457** | **South Korea** | **Present study** |
| 3 | *P. peninsularis* | | D-loop | NC063096 | South Korea | [37] |
| 4 | *P. peninsularis* | FLMNH10952 | D-loop | GQ395743 | USA | [38] |
| 5 | *P. peninsularis* | FLMNH150993 | D-loop | GQ395739 | USA | [38] |
| 6 | *P. peninsularis* | HBS108722 | D-loop | KC687239 | USA | [30] |
| 7 | *P. peninsularis* | FTA821 | D-loop | KC687237 | USA | [30] |
| 8 | *P. peninsularis* | FLMNH151505 | D-loop | KC687236 | USA | [30] |
| 9 | *P. peninsularis* | CRI3953 | D-loop | KC687235 | USA | [30] |
| 10 | *P. concinna* | HBS123561 | D-loop | KC687215 | USA | [30] |
| 11 | *P. concinna* | HBS123555 | D-loop | KC687213 | USA | [30] |
| 12 | *P. concinna* | HBS117819 | D-loop | KC687212 | USA | [30] |
| 13 | *P. nelsoni* | HBS108599 | D-loop | KC687233 | USA | [30] |
| 14 | *P. nelsoni* | FTA832 | D-loop | KC687232 | USA | [30] |
| 15 | *P. nelsoni* | MVZ 164993 | D-loop | MT424571 | USA | Papenfuss and Stuart [Unpublished] |
| 16 | *P. alabamensis* | HBS123564 | D-loop | KC687171 | USA | [30] |
| 17 | *P. alabamensis* | HBS123559 | D-loop | KC687170 | USA | [30] |
| 18 | *P. alabamensis* | HBS123557 | D-loop | KC687169 | USA | [30] |
| 19 | *P. rubriventris* | USNM544373 | D-loop | KC687243 | USA | [30] |
| 20 | *P. rubriventris* | MCZ188655 | D-loop | KC687242 | USA | [30] |
| 21 | *P. rubriventris* | MCZ188654 | D-loop | KC687241 | USA | [30] |
| 22 | *P. texana* | RCT2 | D-loop | KX558427 | USA | [30] |
| 23 | *P. texana* | RCT66 | D-loop | KC687247 | USA | [30] |
| 24 | *P. texana* | RCT2 | D-loop | KC687245 | USA | [30] |
| 25 | *Chrysemys picta* | | D-loop | JN993967 | USA | Myers [Unpublished] |

The bold fonts indicate *de novo* sequences obtained from *Pseudemys peninsularis* eggshells in the present study.

generations. The convergence was judged in the software TRACER version 1.6 [34] using the average standard deviation of split frequencies ($< 0.01$) and the ESS values ($> 200$). We used four Markov Chain Monte Carlo chains to perform two independent runs and discarded 25% of the tree as burn-in. Finally, the obtained tree was visualized with the help of FigTree 1.4.2 [35]. The ML tree was constructed through RAxML 7.0.4 [36], with the GTR substitution model, gamma-distributed rate heterogeneity among sites, and the invariable proportion sites estimated from the data. The analysis used 1000 bootstrap pseudo-replicates. We considered the nodes with equal or more than 0.95 Bayesian Posterior Probabilities (BPP) in the BI tree and equal or more than 70% Bootstrap Support (BS) in the ML tree as well supported. Furthermore, genetic distances were calculated under the uncorrected p-distance model using MEGA X [31].

## Artificial incubation and identification of the hatchlings

The collected eggs were incubated at 28–30°C. This temperature range is proven to give a high success rate in a short incubation period [39]. The hydrated moss was placed underneath, which was replaced every two weeks, to maintain the humidity. The eggs were hatched in 47–50 days. The hatchlings were identified through external morphological features.

### Ethical permit

This study was approved by the Yeongsangang River Basin Environmental Office of the Korea Ministry of Environment (permission number: 2021–8). The experiments were conducted following the instructions and ethical permissions from the ethical committee, Laboratory Animal Research Center, Chonnam National University, South Korea (Certificate No. 2020–154).

## Result

### Phylogenetic analysis

The sequences from our study clustered with other sequences of *P. peninsularis* from the GenBank in the phylogenetic tree. We found *P. peninsularis* to be a sister clade to *P. nelsoni*. Whereas, *P. rubiventris* was a sister clade to *P. peninsularis*, *P. nelsoni*, *P. concinna*, *P. albamensis*, and *P. texana*. However, the taxa of the latter clade were highly inclusive with strong node supports (Fig 2A). Furthermore, the BLAST results also revealed similar patterns to phylogenetic analysis. BLAST in NCBI showed the generated sequences had a 99.80% similarity with the sequences of *P. peninsularis* deposited in the GenBank. The *de novo* sequences showed very low genetic distances with the sequences of the same species from the GenBank than other species (Fig 2B). Our sequences had 0% to 1% uncorrected *p*-distances with other sequences of *P. peninsularis*.

### Nest and egg characteristics

We discovered two nests of *P. peninsularis* at the study area. The clutch size of the two nests was 27 and 07 respectively. As 10 eggs were damaged during the collection, we measured 24 eggs for the identifications. The eggs were oval-elongated shaped and white in color. The eggshells were of parchment type (Fig 1C). We found a length of 31.43mm (±0.878), a width of 22.62mm (±0.549), an elongation of 1.39 (±0.067), a mass of 9.15g (±0.607). In addition, the compilation of egg properties of eight freshwater turtle species from South Korea revealed differences in species level except for *Trachemy scripta* and *Graptemys ouachitensis* (Fig 3). These two species had the same egg elongation of 1.64, while others had different egg elongations and thus different shapes (Table 2).

### Hatchling characteristics

The carapace of hatchlings was dark green with parallel yellow stripes. The marginal shells were with a marginal yellow line. There was a green-colored mid-dorsal keel. The dorsal side legs, tail, and neck were mostly dark green with longitudinal yellow stripes. The yellow stripes on the head form a U-shape behind the eyes. The plastron was unhinged and mostly yellow in color (Fig 2C and 2D).

## Discussion

We found two *P. peninsularis* turtle nests in Jeonpyeongje Neighborhood Park, Maewol-dong, Seo-gu, Gwangju, South Korea (Fig 1B–1D), and successfully identified them by combining the egg characteristics, morphological features of the hatchlings, and phylogenetic analysis of partial mitochondrial DNA from the eggshells. This was the first initiative to extract DNA from eggshells, which is already well-practiced in sea turtles [25, 26]. Although our initiative was successful, extracting DNA from eggshells is challenging as it is expected to get a lower concentration than that may yield from muscle or blood [40]. The low concentration and poor template quality may lead to amplification failure [41, 42]. However, using DNA from eggshells is convenient, especially, to identify the nests because it does not require the individual

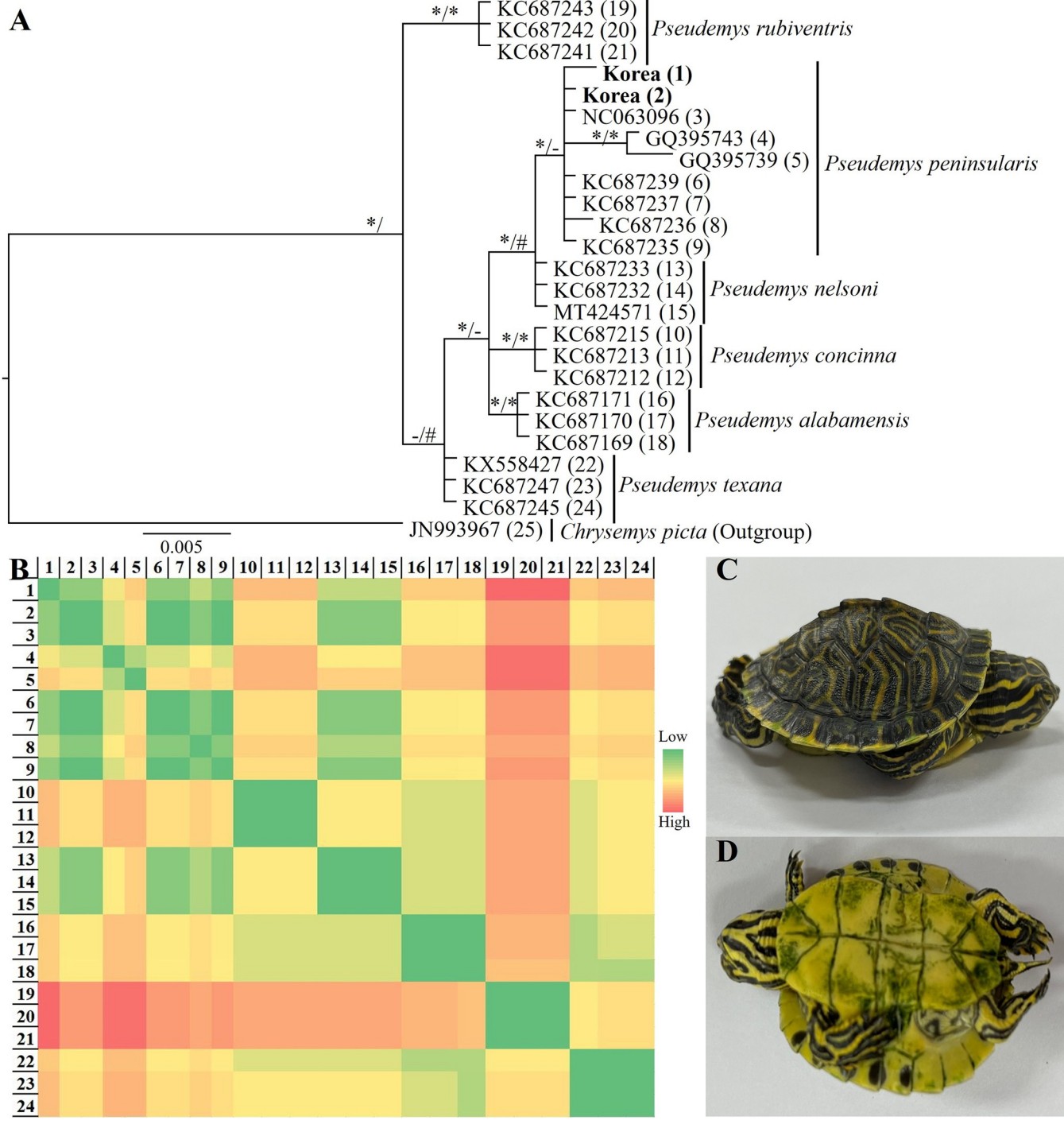

**Fig 2. Identification of the samples.** (A) Phylogenetic tree based on mitochondrial DNA control region. Nodal support values placed before '/' are Bayesian Posterior Probabilities (BPP), and values placed after '/' are Maximum Likelihood Bootstraps (BS). In our tree '*' denotes greater or equal to 0.95 BPP and 70% BS, '#' denotes 0.85–0.94 BPP and 60–69% BS, and '–' denotes below 0.85 BPP and 60% BS. The numbers in the parentheses correspond to the serial number of Table 1. The bold fonts indicate the *de novo* sequences. (B) Pairwise genetic distance (*p*-distance). We excluded the outgroup from the calculation. The numbers correspond to the serial number of Table 1. (C) Dorso-lateral view of a hatchling of *Pseudemys peninsularis*. (D) Ventral view of a hatchling of *P. peninsularis*.

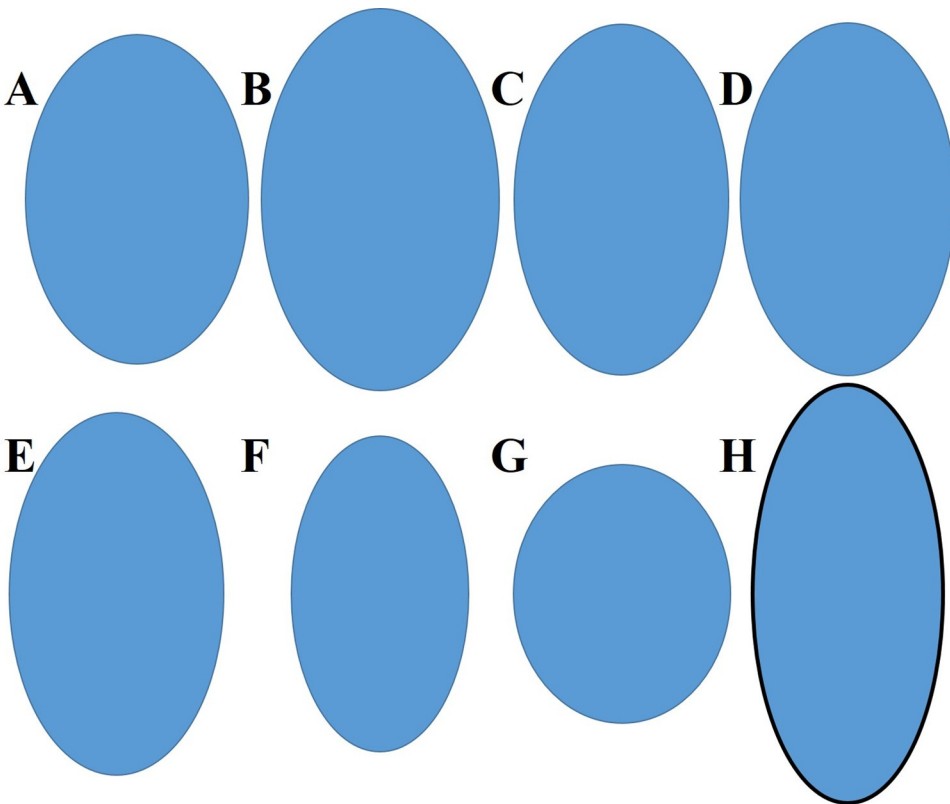

**Fig 3. Egg shapes and eggshells of freshwater turtles found in South Korea.** (A) *Pseudemys peninsularis*; (B) *P. concinna*; (C) *Trachemys scripta*; (D) *Graptemys ouachitensis*; (E) *G. geographica*; (F) *Chrysemys picta*; (G) *Chelydra serpentina*; (H) *Mauremys reevesii*. The eggs without outlines indicate parchment shells and the egg with a black outline indicates a brittle shell.

turtles [26, 43]. Furthermore, freshwater turtle nests are difficult to find and are also difficult and time-consuming to hatch the eggs to identify the nests [15]. Thus, although *P. peninsularis* is widely distributed in South Korea and in many other non-native countries, there are no reports on nests of this species in nature [27]. This is the first report of *P. peninsularis* nests in nature outside its native range [15].

However, the BLAST results and phylogenetic analysis of the partial mitochondrial DNA from our samples revealed a close relationship and low genetic distances between *P. peninsularis* and *P. nelsoni*, *P. concinna*, and *P. albamensis* (Fig 2B). This finding is similar to the previous studies [30, 38]. The BLAST results suggested a high similarity between the novel sequences of *P. penisularis* and *P. nelsoni* (99.59%), *P. concinna* (99.39%), and *P. alabamensis* (98.99%). Similarly, our sequences showed low genetic distances with these species. The uncorrected *p*-distances between our sequences and *P. nelsoni*, *P. concinna*, *P. alabamensis*, *P. texana*, and *P. rubiventris* were 0.2–0.4%, 0.8% - 1.2%, 0.7–1.0%, 0.8%– 1.2%, and 1.6%– 2.2% respectively. The phylogenetic tree topology also supported the BLAST results and uncorrected *p*-distances by forming inclusive clades with strong node supports (Fig 2A). Although the clades of *P. peninsularis*, *P. nelsoni*, *P. concinna*, and *P. alabamensis* were inclusive and only *P. rubiventris* was a sister clade to them, our results recovered almost similar topology as previous genetic studies on the *Pseudemys* genus and supported the hypothesis of taxonomic confusions in this genus [30, 38].

**Table 2. Egg characteristics of eight freshwater turtles from South Korea.**

| Species | | A | B | C | D | E | F | G | H |
|---|---|---|---|---|---|---|---|---|---|
| Shell type | | α | α | α | α | α | α | α | β |
| Mass (g) | i | 9.15 | | | | | | | |
| | ii | | 15.44 | 9.74 | 10.06 | 10.57 | 6.67 | 9.91 | 10.17 |
| | iii | 11.49 | 11.99 | 10.52 | | | 6.17 | 9.63 | |
| | Mean | 10.32 | 13.72 | 10.13 | 10.06 | 10.57 | 6.42 | 9.77 | 10.17 |
| Length (mm) | i | 31.43 | | | | | | | |
| | ii | | 42.03 | 35.63 | 36.18 | 37.20 | 31.56 | 26.49 | 42.88 |
| | iii | 36.22 | 36.36 | 36.39 | | | 33.22 | | |
| | Mean | 33.83 | 39.20 | 36.01 | 36.18 | 37.20 | 32.39 | 26.49 | 42.88 |
| Width (mm) | i | 22.62 | | | | | | | |
| | ii | | 24.92 | 21.83 | 22.08 | 22.04 | 19.10 | 25.25 | 19.47 |
| | iii | 23.15 | 23.90 | 22.17 | | | 17.28 | 25.78* | |
| | Mean | 22.89 | 24.41 | 22.00 | 22.08 | 22.04 | 18.19 | 25.25 | 19.47 |
| Elongation | | 1.48 | 1.61 | 1.64 | 1.64 | 1.69 | 1.78 | 1.05 | 2.20 |

A = *Pseudemys peninsularis*; B = *P. concinna*; C = *Trachemys scripta*; D = *Graptemys ouachitensis*; E = *G. geographica*; F = *Chrysemys picta*; G = *Chelydra serpentina*; H = *Mauremys reevesii*; α = Parchment; β = Brittle.

* = does not have the egg length in the corresponding literature and is thus excluded from the calculation. i = Present study; ii = Congdon and Gibbons (1985) [28];
iii = Iverson and Ewert (1991) [18].

In addition to molecular and morphology-based species identifications, egg properties also can be used in turtle nest identification. Especially, egg elongation is useful in identifying turtle species [18, 19]. We got an egg elongation of 1.48 for *P. penisularis* in the present study, which is similar to the previous studies [28]. Additionally, the compilation of data on egg properties of eight turtle species from South Korea revealed an egg elongation of 1.61, 1.69, 1.78, 1.05, and 2.20 for *P. concinna, G. geographica, C. picta,* and *M. reevesii* respectively (Table 2). The other two species, *T. scripta* and *G. ouachitensis* had the same egg elongation, 1.64. Furthermore, we recovered the schematic diagram of the egg shapes based on the present and previous studies (Fig 3) [18, 28], which showed the effectiveness of using egg properties in identifying turtle nests.

The identification and discovery of freshwater turtle nests have always been very difficult for researchers [15]. Thus, the nests of *P. peninsularis* have not been reported from outside of its native range even after being widely used as a pet species and having a wide distribution in America, Europe, and Asia (Fig 1A) [44–48]. In South Korea, this species has been reported from all over the country including Jeju island [27]. Considering the large body size and capability of basking in a high temperature [49, 50], this species has the potential to pose competition pressure and threats to native species. In addition to proven negative impacts on other turtle species [51–53], this species has been reported to bear parasites like *Aeromonas enteropelogenes, A. hydrophila,* and *A. veronii,* potential to affect amphibian and human health [54]. Still, this species has not been designated as an ecosystem-disturbing species as it lacks confirmed information on its successful establishment and reproduction [12], which can be evident by reports on its nests in the local environment [13, 14].

The present study has confirmed the successful establishment and reproduction of *P. peninsularis* in South Korea by identifying its nests in the local environment through traditional and molecular approaches. Thus, considering the distribution and potential negative impact of this alien species on native ecosystems and human beings, we urge immediate designation of *P. peninsularis* as an ecosystem-disturbing species and its control and management policies. In

addition, our study is highly significant as it included comparative descriptions and schematic diagrams of the eggs of eight freshwater turtles from South Korea. It will help scientists in studying freshwater turtles, including alien and also invasive species, in this country. Furthermore, we have presented the first-ever successful use of eggshell DNA in identifying freshwater turtles. We believe it could be significant tool for future researches in identifying the alien invasive freshwater turtle nests and developing their control and management policies across the world.

## Author Contributions

**Conceptualization:** Md. Mizanur Rahman, Dong-Hyun Lee, Ha-Cheol Sung.

**Data curation:** Md. Mizanur Rahman.

**Formal analysis:** Md. Mizanur Rahman.

**Funding acquisition:** Ha-Cheol Sung.

**Investigation:** Seung-Ju Cheon, Ji-A Lee, Seung-Min Park, Jae-Hong Park.

**Methodology:** Seung-Ju Cheon, Md. Mizanur Rahman, Ha-Cheol Sung.

**Software:** Md. Mizanur Rahman.

**Supervision:** Md. Mizanur Rahman, Dong-Hyun Lee, Ha-Cheol Sung.

**Validation:** Md. Mizanur Rahman.

**Visualization:** Seung-Ju Cheon, Md. Mizanur Rahman.

**Writing – original draft:** Seung-Ju Cheon, Md. Mizanur Rahman.

**Writing – review & editing:** Seung-Ju Cheon, Md. Mizanur Rahman, Ji-A Lee, Seung-Min Park, Jae-Hong Park, Dong-Hyun Lee, Ha-Cheol Sung.

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
