## [Decision Letter · Decision Letter 0]

19 Dec 2022

PONE-D-22-32805Confirmation of the local establishment of alien invasive turtle, Pseudemys peninsularis, in South Korea using eggshell DNAPLOS ONE

Dear Dr. Sung,

Thank you for submitting your manuscript to PLOS ONE. After careful consideration, we feel that it has merit but does not fully meet PLOS ONE’s publication criteria as it currently stands. Therefore, we invite you to submit a revised version of the manuscript that addresses the points raised during the review process.

We look forward to receiving your revised manuscript.

Kind regards,

Tzen-Yuh Chiang

Academic Editor

PLOS ONE

Journal Requirements:

"This work was supported by Korea Environment Industry & Technology Institute (KEITI) through the Project for the Development of Biological Diversity Threats Outbreak Management Technology (RE201807039), funded by the Korea Ministry of Environment (MOE). "

"H-CS: RE201807039; Korea Environment Industry & Technology Institute.

4. We note that Figures 1a,1b and 1d in your submission contain [map/satellite] images which may be copyrighted. All PLOS content is published under the Creative Commons Attribution License (CC BY 4.0), which means that the manuscript, images, and Supporting Information files will be freely available online, and any third party is permitted to access, download, copy, distribute, and use these materials in any way, even commercially, with proper attribution. For these reasons, we cannot publish previously copyrighted maps or satellite images created using proprietary data, such as Google software (Google Maps, Street View, and Earth). For more information, see our copyright guidelines: http://journals.plos.org/plosone/s/licenses-and-copyright.

a. You may seek permission from the original copyright holder of Figures 1a,1b and 1d to publish the content specifically under the CC BY 4.0 license.  

5. We note that you have referenced Unpublished in page 6 which has currently not yet been accepted for publication. Please remove this from your References and amend this to state in the body of your manuscript: (ie “Bewick et al. [Unpublished]”) as detailed online in our guide for authors

Reviewers' comments:

Reviewer's Responses to Questions

**Comments to the Author**

1. Is the manuscript technically sound, and do the data support the conclusions?

Reviewer #1: Yes

2. Has the statistical analysis been performed appropriately and rigorously? 

Reviewer #1: Yes

3. Have the authors made all data underlying the findings in their manuscript fully available?

Reviewer #1: Yes

4. Is the manuscript presented in an intelligible fashion and written in standard English?

Reviewer #1: Yes

5. Review Comments to the Author

Reviewer #1: Dear Authors,

I read the MS titled "Confirmation of the local establishment of alien invasive turtle, Pseudemys peninsularis, in South Korea using eggshell DNA" with interest. Congratulations on a catchy article. I see the merit of the given study for further monitoring and research activities and it is obvious that the goal of the paper should be overlapped to future practice widely. The text is clearly written, and the analyses and outputs are well-presented and described. Just a few mine minor comments are highlighted directly in the PDF copy attached. Check the format of the listed references carefully. The comments and suggestions are not plenty and easy to follow. After incorporating the suggestions in the text, it is my great pleasure to recommend this MS for publication in PLOS ONE.

6. PLOS authors have the option to publish the peer review history of their article (what does this mean?). If published, this will include your full peer review and any attached files.

Reviewer #1: No

---

## [Author Response · Author response to Decision Letter 0]

19 Jan 2023

Point-by-point response to academic editor and reviewer’s comments

Response to comments from the academic editor

Comment 1: Please ensure that your manuscript meets PLOS ONE's style requirements, including those for file naming. The PLOS ONE style templates can be found at 

Response: Thanks for your comment. We have revised the manuscript accordingly. We have modified the symbols of the author’s affiliation, font sizes of headings and subheadings, spacing of the first lines of all paragraphs, formatted tables, etc., following the guidelines.

Comment 2: Thank you for stating the following in the Acknowledgments Section of your manuscript: 

"This work was supported by Korea Environment Industry & Technology Institute (KEITI) through the Project for the Development of Biological Diversity Threats Outbreak Management Technology (RE201807039), funded by the Korea Ministry of Environment (MOE). "

"H-CS: RE201807039; Korea Environment Industry & Technology Institute. The funders had no role in study design, data collection and analysis, decision to publish, or preparation of the manuscript."

Response: We appreciate your suggestion. We have deleted the ‘Acknowledgements’ section and omitted funding information in the revised manuscript. We have included the amended 'Funding Statement' in the cover letter.

Comment 3: In your Data Availability statement, you have not specified where the minimal data set underlying the results described in your manuscript can be found. PLOS defines a study's minimal data set as the underlying data used to reach the conclusions drawn in the manuscript and any additional data required to replicate the reported study findings in their entirety. All PLOS journals require that the minimal data set be made fully available. For more information about our data policy, please see http://journals.plos.org/plosone/s/data-availability.

Response: We highly appreciate your comment. We have modified the ‘Data Availability Statement’ accordingly. All relevant data are within the manuscript. Please, check line 287.

Comment 4: We note that Figures 1a,1b and 1d in your submission contain [map/satellite] images which may be copyrighted. All PLOS content is published under the Creative Commons Attribution License (CC BY 4.0), which means that the manuscript, images, and Supporting Information files will be freely available online, and any third party is permitted to access, download, copy, distribute, and use these materials in any way, even commercially, with proper attribution. For these reasons, we cannot publish previously copyrighted maps or satellite images created using proprietary data, such as Google software (Google Maps, Street View, and Earth). For more information, see our copyright guidelines: http://journals.plos.org/plosone/s/licenses-and-copyright.

a. You may seek permission from the original copyright holder of Figures 1a,1b and 1d to publish the content specifically under the CC BY 4.0 license. 

Response: We appreciate your concern. The maps and satellite images were generated using Arcmap 10.8.1 (ESRI, USA; https://support.esri.com/en/products/desktop/arcgis-desktop/arcmap/10-8-1), which is free to use and we referred the source in the figure caption. Please, check lines 108 – 113 of the revised manuscript.

Comment 5: We note that you have referenced Unpublished in page 6 which has currently not yet been accepted for publication. Please remove this from your References and amend this to state in the body of your manuscript: (ie “Bewick et al. [Unpublished]”) as detailed online in our guide for authors

Response: Thanks for your advice. We have changed the reference following your suggestion. Please, check Table 1 of the revised manuscript. 

Comment 6: Please review your reference list to ensure that it is complete and correct. If you have cited papers that have been retracted, please include the rationale for doing so in the manuscript text, or remove these references and replace them with relevant current references. Any changes to the reference list should be mentioned in the rebuttal letter that accompanies your revised manuscript. If you need to cite a retracted article, indicate the article’s retracted status in the References list and also include a citation and full reference for the retraction notice.

Response: We appreciate your suggestion. We have checked the references and confirmed their complete and correctness. Besides, we have added one additional reference suggested by the reviewer. Please, check lines 314 – 316. We have mentioned that in the response of reviewer’s comments too.

Response to comments from the Reviewer: 

Comment 1: I read the MS titled "Confirmation of the local establishment of alien invasive turtle, Pseudemys peninsularis, in South Korea using eggshell DNA" with interest. Congratulations on a catchy article. I see the merit of the given study for further monitoring and research activities and it is obvious that the goal of the paper should be overlapped to future practice widely. The text is clearly written, and the analyses and outputs are well-presented and described. Just a few mine minor comments are highlighted directly in the PDF copy attached. Check the format of the listed references carefully. The comments and suggestions are not plenty and easy to follow. After incorporating the suggestions in the text, it is my great pleasure to recommend this MS for publication in PLOS ONE.

Response: We highly appreciate your reviewing efforts and positive comments. We have modified the manuscript as per your instructions and tried to resolve all issues you mentioned. Herein, we are responding to each of your concerns point by point.

Comment 2: ‘Not only, un wanted and undesired pets are also released’

Response: Thanks for your comment. We have added the information in our revised manuscript. Please, check line 27 – 28.

Comment 3: ‘I suggest to add a brief note on the focused legislative regulations which are unfortunately ineffective in many cases. See: Patoka, J., Magalhães, A. L. B., Kouba, A., Faulkes, Z., Jerikho, R., & Vitule, J. R. S. (2018). Invasive aquatic pets: failed policies increase risks of harmful invasions. Biodiversity and Conservation, 27(11), 3037-3046.’

Response: We highly appreciate your advice. We have added the suggested brief note in the updated manuscript. Please, check lines 50 – 54.

Comment 4: ‘Do you know how many turtle species are pet-trade in the country?’

Response: Thanks for your question. There are 110 turtle species are pet-trade in the country. We also have included the information in the manuscript. Please, check line 57 of the revised manuscript.

Comment 5: ‘This info has to be moved to the results section.’

Response: We appreciate your suggestion. We have transferred it to the results section as per your suggestion. Please, check line 194 of the manuscript.

Comment 6: ‘Since the captions have to be self-explaining, the full name is required in each caption where is the first mentioned.’

Response: Thanks for your valuable comment. We have modified the scientific name accordingly. Please, check lines 155 – 156.

Comment 7: ‘Full name in this cell’

Response: Thanks for your advice. We have modified it accordingly. Please, check Table 1.

Comment 8: ‘Unpublished’

Response: We highly appreciate your comment. We have corrected the word accordingly. Please, check Table 1.

Comment 9: ‘The full name here’

Response: Thanks for the suggestion. We have modified the scientific name. Please, check line 190 – 191 of the revised manuscript.

Comment 10: ‘Italicized’

Response: Thanks for the advice. We have modified the scientific name. Please, check line 191 of the revised manuscript.

Comment 11: ‘Italicized’

Response: Thanks for the suggestion. We have modified the scientific name. Please, check line 205 of the revised manuscript.

Comment 12: ‘abbreviated Graptemys’

Response: We appreciate your suggestion. We have abbreviated the scientific name. Please, check line 206 of the revised manuscript.

Comment 13: ‘Here, P. concinna is adequate’

Response: Thanks for the advice. We have modified the scientific name. Please, check line 211 of the revised manuscript.

Comment 14: ‘abbreviated Graptemys’

Response: Thanks for the suggestion. We have modified the scientific name. Please, check line 211 – 212 of the revised manuscript.

Comment 15: ‘Italicized’

Response: We appreciate your advice. We have italicized the scientific name. Please, check line 254 of the revised manuscript.

Comment 16: ‘Check all these species if cited above or not. If yes, use the genus abbreviation.’

Response: Thanks for the advice. We have checked and modified the scientific names accordingly. Please, check lines 257 – 258 of the revised manuscript.

Comment 17: ‘Change i to y.’

Response: Thanks for the comment. We have rechecked the whole manuscript and corrected the scientific names if needed. Herein, we have abbreviated the genus name following your previous suggestion. Please, check line 257 of the revised manuscript.

Comment 18: ‘including alien and also invasive species.’

Response: We appreciate your suggestion. We have modified the sentence accordingly. Please, check lines 280 – 281 of the revised manuscript.

Comment 19: ‘Change m to M.’

Response: We highly appreciate your comment. We have modified the name accordingly. Please, check line 292 of the manuscript.

Comment 20: ‘Change y to Y.’

Response: Thanks for the advice. We have corrected it accordingly. Please, check line 302of the revised manuscript.

Comment 21: ‘Italicized.’

Response: We appreciate your suggestion. We have corrected the scientific name and turned it into italicized. Please, check line 308 of the revised manuscript.

Comment 22: ‘Family, thus regular and not italicized.’

Response: Thanks for the comment. We have turned it in to non-italicized regular font. Please, check line 308 of the revised manuscript.

Comment 23: ‘Just article No. 12:13143.’

Response: We appreciate your comment. We have corrected accordingly. Please, check line 315 of the revised manuscript.

Comment 24: ‘- or –?’

Response: We highly appreciate your comment. We have corrected accordingly. Please, check line 319 of the revised manuscript.

Comment 25: ‘dtto.’

Response: Thanks for your comment. We have corrected accordingly. Please, check line 327 of the revised manuscript.

Comment 26: ‘Change c to C and b to B.’

Response: We appreciate your advice. We have corrected accordingly. Please, check line 349 of the revised manuscript.

Comment 27: ‘Why abbreviated.’

Response: We highly appreciate your comment. We have corrected accordingly. Please, check lines 410 – 411 of the revised manuscript.

Comment 28: ‘Just article No. 410:2.’

Response: Thanks for your suggestion. We have corrected accordingly. Please, check line 435 of the revised manuscript.

Comment 29: ‘Aquatic Organisms.’

Response: We appreciate your advice. We have corrected accordingly. Please, check line 453 of the revised manuscript.

Comment 30: ‘Italicized.’

Response: We appreciate your comment. We have corrected accordingly. Please, check line 460 of the revised manuscript.

---

## [Decision Letter · Decision Letter 1]

2 Feb 2023

Confirmation of the local establishment of alien invasive turtle, Pseudemys peninsularis, in South Korea, using eggshell DNA

PONE-D-22-32805R1

Dear Dr. Sung,

We’re pleased to inform you that your manuscript has been judged scientifically suitable for publication and will be formally accepted for publication once it meets all outstanding technical requirements.

Kind regards,

Tzen-Yuh Chiang

Academic Editor

PLOS ONE

Additional Editor Comments (optional):

Reviewers' comments:

Reviewer's Responses to Questions

**Comments to the Author**

1. If the authors have adequately addressed your comments raised in a previous round of review and you feel that this manuscript is now acceptable for publication, you may indicate that here to bypass the “Comments to the Author” section, enter your conflict of interest statement in the “Confidential to Editor” section, and submit your "Accept" recommendation.

Reviewer #1: All comments have been addressed

2. Is the manuscript technically sound, and do the data support the conclusions?

Reviewer #1: Yes

3. Has the statistical analysis been performed appropriately and rigorously? 

Reviewer #1: N/A

4. Have the authors made all data underlying the findings in their manuscript fully available?

Reviewer #1: Yes

5. Is the manuscript presented in an intelligible fashion and written in standard English?

Reviewer #1: Yes

6. Review Comments to the Author

Reviewer #1: Dear Authors,

I am satisfied with the revised version of your manuscript. The text was improved and all my comments and suggestions were incorporated into the text. Just correct the name of P. nelsoni in the Table 1. I have no further comments and congratulations to the catchy study.

Sincerely

7. PLOS authors have the option to publish the peer review history of their article (what does this mean?). If published, this will include your full peer review and any attached files.

Reviewer #1: **Yes: **Jiří Patoka

---

## [Editor Report · Acceptance letter]

7 Feb 2023

PONE-D-22-32805R1 

Confirmation of the local establishment of alien invasive turtle, *Pseudemys peninsularis*, in South Korea, using eggshell DNA 

Dear Dr. Sung:

I'm pleased to inform you that your manuscript has been deemed suitable for publication in PLOS ONE. Congratulations! Your manuscript is now with our production department. 

Kind regards, 

on behalf of

Dr. Tzen-Yuh Chiang 

Academic Editor

PLOS ONE